# Investigation of the Uptake and Transport of Two Novel Camptothecin Derivatives in Caco-2 Cell Monolayers

**DOI:** 10.3390/molecules27123669

**Published:** 2022-06-07

**Authors:** Yi Wang, Xiangli Zhang, Wenya Zhuang, Yanlei Yu, Xuanrong Sun, Hong Wang, Fengzhi Li, Qingyong Li

**Affiliations:** 1College of Pharmaceutical Science, Zhejiang University of Technology, Hangzhou 310032, China; 2111907037@zjut.edu.cn (Y.W.); zhangxl_07@126.com (X.Z.); 2112007126@zjut.edu.cn (W.Z.); yanleiyu@zjut.edu.cn (Y.Y.); sunxr@zjut.edu.cn (X.S.); hongw@zjut.edu.cn (H.W.); 2Department of Pharmacology and Therapeutics, Roswell Park Comprehensive Cancer Center, Buffalo, NY 14263, USA; fengzhi.li@roswellpark.org

**Keywords:** Camptothecin derivatives, cytotoxicity, bioavailability, Caco-2 cells model, drug resistance

## Abstract

Irinotecan and Topotecan are two Camptothecin derivatives (CPTs) whose resistance is associated with the high expression of breast cancer resistance protein (BCRP) and P-glycoprotein (P-gp). To reverse this resistance, two novel CPTs, FL77-28 (7-(3-Fluoro-4-methylphenyl)-10,11-methylenedioxy-20(S)-CPT) and FL77-29 (7-(4-Fluoro-3-methylphenyl)-10,11-methylenedioxy-20(S)-CPT), were synthesized by our group. In this study, the anti-tumor activities of FL77-28, FL77-29, and their parent, FL118 (10,11-methylenedioxy-20(S)-CPT), were evaluated and the results showed that FL77-28 and FL77-29 had stronger anti-tumor activities than FL118. The transport and uptake of FL118, FL77-28, and FL77-29 were investigated in Caco-2 cells for the preliminary prediction of intestinal absorption. The apparent permeability coefficient from apical to basolateral (P_app AP-BL_) values of FL77-28 and FL77-29 were (2.32 ± 0.04) × 10^−6^ cm/s and (2.48 ± 0.18) × 10^−6^ cm/s, respectively, suggesting that the compounds had moderate absorption. Since the transport property of FL77-28 was passive diffusion and the efflux ratio (ER) was less than 2, two chemical inhibitors were added to further confirm the involvement of efflux proteins. The results showed that FL77-28 was not a substrate of P-gp or BCRP, but FL77-29 was mediated by P-gp. In conclusion, FL77-28 might be a promising candidate to overcome drug resistance induced by multiple efflux proteins.

## 1. Introduction

The therapeutic effectiveness of drugs highly depends upon their bioavailability. Intestinal absorption, a key factor affecting the bioavailability of orally administered drugs, is usually affected by various ATP-binding cassette transporters (ABC transporters). BCRP and P-gp are major drug efflux pumps in the ABC transporters, which are closely related to drug resistance [1,2,3]. The presence of P-gp and BCRP in epithelial cells of the gastrointestinal tract could restrict absorption and thus the oral bioavailability of their substrate [4]. Therefore, inhibition of the activity of P-gp and BCRP may reduce the efflux of various drugs that are their substrates, thereby improving drug efficacy and bioavailability. Multidrug resistance (MDR) to many anticancer drugs that served as substrates (e.g., taxanes, vinca alkaloids, anthracyclines, and epipodophyllotoxin) has been reported to be closely associated with ABC transporters [5,6]. Nowadays, compounds derived from natural products have been candidates for the effective reversal of MDR [7].

Camptothecin (CPT), a monoterpenoid quinoline alkaloid isolated from *Camptotheca acuminate*, shows a broad-spectrum anti-tumor activity, but its clinical application is limited by its poor water solubility, unstable lactone ring, low bioavailability, and severe toxic side effects [8,9]. Therefore, a series of CPTs were synthesized through structural modification, hoping to enhance efficacy and decrease toxicity. As water-soluble CPTs, Irinotecan and Topotecan, approved by the FDA for clinical cancer treatment, showed superior antitumor activity and low toxicity [10,11]. Unfortunately, drug resistance against Irinotecan and Topotecan has emerged in clinical treatment. The intracellular accumulation of Topotecan, Irinotecan, and Irinotecan active metabolite SN-38 as substrates of BCRP and P-gp was reduced due to the overexpression of these efflux proteins [12,13,14]. In 2012, FL118, a C10,11-methylenedioxy substituted CPT, was rediscovered via high-throughput screening [15]. Subsequent studies showed that FL118 possessed superior antitumor activity to Topotecan and Irinotecan, while not being a substrate for P-gp or BCRP, suggesting that FL118 seems to avoid the drug resistance caused by efflux proteins [16,17,18].

According to published structure–activity relationship (SAR) studies, modifications at the C7-position of Camptothecin appears to be one of the most effective ways to improve anti-tumor potency [19,20]. Furthermore, several reports documented that adding lipophilic groups at the C7 position was beneficial to improve the lipophilicity of CPT and prolong its retention time in vivo [21,22]. Based on the above background, two new lipid-soluble derivatives, FL77-28 and FL77-29, were obtained through optimizing the C7 substitution of FL118 to further improve anti-tumor activity and intestinal absorption.

In this study, the absorption mechanisms and permeability of FL77-28 and FL77-29 were explored. In addition, the properties of FL118, FL77-28, and FL77-29 to overcome P-gp and BCRP-induced resistance and the effect of cellular uptake at different drug concentrations, incubation temperatures, and transporter inhibitors were also investigated.

## 2. Materials and Methods

### 2.1. Chemicals and Materials

FL118, FL77-28, and FL77-29 (purity > 99%) were synthesized by our lab. Verapamil Hydrochloride, Gefitinib, and bovine serum albumin (BSA) were obtained from Aladdin (Shanghai, China). Dulbecco’s modified Eagle’s medium (DMEM) was obtained from Gibco BRL Life Technology (Grand Island, NY, USA). Fetal bovine serum (FBS) was obtained from Hyclone (Shanghai, China). Methyl thiazolyl tetrazolium (MTT) and 0.25% trypsin-EDTA were purchased from Solarbio (Beijing, China). Phosphate-buffered saline (PBS), Hank’s buffered salt solution (HBSS) containing 10 mM N-2-hydroxyethyl piperazine-N′-2-ethane sulfonic acid (HEPES), and 15 mM glucose were purchased from Cienry (Huzhou, China). Fasted state simulated intestinal fluid (FaSSIF) was obtained from Biorelevant (London, UK). DMSO and Lucifer yellow (LY) were obtained from Sigma-Aldrich (Shanghai, China). Transwell^®^ permeable support (0.4 μm pore size polycarbonate membrane and 12 mm diameter) was purchased from Corning-Costar (Shanghai, China).

### 2.2. Cell Culture

HCT 116, HeLa, A549, Hep G2, and Caco-2 cell lines were obtained from the China Center for Type Culture Collection, CCTCC (Wuhan, China). They were cultured in DMEM, including 10% FBS, 1% penicillin (l00 μg/mL), and 1% streptomycin (l00 μg/mL), in a constant temperature incubator at 37 °C with 5% CO_2_.

### 2.3. Cytotoxicity Assay

The cytotoxicity of FL118, FL77-28, and FL77-29 against HCT116, HeLa, A549 HepG2, and Caco-2 cell lines were measured using MTT assays [23]. The cells were plated into 96-well plates at a density of 2 × 10^4^ cells/mL and cultured for 24 h. Next, the cells were treated with 0.0125–0.500 μM of FL118, FL77-28, and FL77-29 for 4 h or 72 h. After incubation with 20 µL of 5 mg/mL MTT solution for 4 h, the supernatant was gently discarded and 150 µL of DMSO was added to dissolve the formazan crystals. Finally, the absorbance was measured at 570 nm and the background was subtracted at 630 nm using an ELISA Plate Reader.

### 2.4. Establishment and Evaluation of Caco-2 Cell Monolayers

When Caco-2 cells reached 80–90% confluence in T-25 flasks, the cells were digested, centrifuged, and resuspended. After seeding at a density of 2 × 10^5^ cells/well onto the apical of the Transwell^®^ plate, 0.5 mL of 10% DMEM was added into the apical (AP) chamber and 1.5 mL of 10% DMEM was added into the basolateral (BL) chamber. The buffer was refreshed every two days, and after one week it was refreshed daily and incubated until 21 days.

The integrity of the Caco-2 cell monolayer was evaluated by measuring the transepithelial electrical resistance (TEER) and the permeability of Lucifer yellow (LY) [24]. The TEER was measured using Millicell^®^ ERS-2 voltameter (Bedford, USA). The Caco-2 cell model was used for the transport of FL77-28 and FL77-29 when the TEER values exceeded 300 Ω/cm^2^. The TEER value was also measured at the end of the transport experiment. In addition, the permeability of LY, a paracellular transport marker, was also measured for the tightness of epithelial cells. Hence, 20 µg/mL of LY was added to the inserts and incubated for 2 h at 37 °C. Then, the fluorescence of samples was measured at excitation/emission wavelengths of 492/520 nm using the ELISA Plate [25].

### 2.5. Transport Studies

Before the transport studies, the Caco-2 cell monolayers were washed twice with warm HBSS containing 15 mM glucose, 10 mM HEPES, and 1% BSA (pH 7.4) and pre-incubated for 30 min at 37 °C [26]. Next, HBSS was discarded and replaced with 0.5/1.0 µM of FL77-28 and FL77-29 dissolved in a different buffer on one side of the cell layer, 0.5 mL for the AP side and 1.5 mL for the BL side. The buffer at the AP was FaSSIF (pH 6.5) and the buffer at the BL was HBSS. Then, the monolayers were incubated at 37 °C. A sample aliquot of 0.5 mL (AP-BL) or 0.25 mL (BL-AP) was taken from the receiving compartment at different time points (120, 150, 180, 210, and 240 min) and then replaced with an equal volume of fresh pre-warmed buffer. The concentrations of FL77-28 and FL77-29 in the sample were determined by HPLC.

### 2.6. Uptake Experiment

Caco-2 cells were plated in a 6-well plate at a density of 2 × 10^5^ cells/well and cultured at 37 °C. After reaching 80% confluence, Caco-2 cells could be used in the uptake experiments. Next, the cells were washed twice with HBSS and pre-incubated with HBSS or 1 µM Verapamil or Gefitinib for 30 min. After pre-incubation, the drug-free HBSS was replaced by FaSSIF containing FL118, FL77-28, or FL77-29 (0.2, 0.4, 0.6, 0.8, and 1.0 μM), and then incubated for 4 h at 37 °C or 4 °C, respectively. Next, the cells were collected and lysed by freezing and thawing three times. Then, the cells were sonicated and centrifuged at 10,000 rpm for 10 min. Finally, the supernatants were used to measure concentrations of compounds and proteins by HPLC and Bradford methods.

### 2.7. HPLC Analysis of Samples

Using the Shimadzu HPLC system (Shimadzu, Kyoto, Japan), the samples taken from the uptake experiment and the transport studies were eluted on a C18 reverse-phase column (250 mm × 4.6 mm, 5 μm, Elite, Dalian). The column was maintained at 40 °C and the absorbance detector wavelength was 365 nm. The mobile phase used for FL118, FL77-28, and FL77-29 consisted of acetonitrile: 0.1% formic acid (40:60, *v*/*v*), (60:40, *v*/*v*), and (65:35, *v*/*v*). The flow rate was 1.0 mL/min, and the injection volume was 20 μL. The retention times of FL118, FL77-28, and FL77-29 were 6.109 min (Figure 1A), 8.456 min (Figure 1B), and 6.351 min (Figure 1C), respectively. Calibration curves were constructed over a concentration range of 0.03–1.00 μM. Linear regression analysis of peak area and concentration revealed a typical equation y = 17807x − 44.08 (R^2^ = 0.9999) for FL118, y = 14629x + 62.94 (R^2^ = 0.9999) for FL77-28, and y = 13541× + 11.433 (R^2^ = 0.9997) for FL77-29. The concentration of FL118, FL77-28, and FL77-29 was determined according to the typical equations.

### 2.8. Data Analysis

The concentration of each sample was measured by HPLC. The P_app_ of FL77-28 and FL77-29 was calculated according to the following equation:P_app_ (cm/s) = V/(C_0_·A)·(ΔQ/Δt)(1)
where V (cm^3^) is the volume of the receiver compartment, A is the monolayer surface area of the Transwell insert (1.12 cm^2^), C_0_ is the initial drug concentration in the donor compartment, and (ΔQ/Δt) is the transport rate.

The efflux ratio (ER) was calculated using the following equation:ER = P_app BL-AP_/P_app AP-BL_(2)

### 2.9. Statistical Analysis

All data were presented as the mean ± SD from three replicate experiments. Statistical analysis was determined using an unpaired Student’s *t*-test. A *p*-value of 0.05 was considered statistically significant.

## 3. Results

### 3.1. Cytotoxicity Test

The in vitro anti-tumor activity of FL118, FL77-28, and FL77-29 on HCT 116, HepG2, A549, and HeLa cell lines was determined by MTT assay. As shown in Table 1, the IC_50_ values of FL77-28 on HCT 116, Hep G2, A549, and HeLa were 0.045 ± 0.014 μM, 0.065 ± 0.001 μM, 0.048 ± 0.007 μM, and 0.022 ± 0.007 μM at 72 h, respectively. Similarly, the IC_50_ value of FL77-29 on HCT 116, Hep G2, A549, and HeLa were 0.025 ± 0.003 μM, 0.034 ± 0.002 μM, 0.028 ± 0.004 μM, and 0.036 ± 0.004 μM at 72 h, respectively. These results revealed that both FL77-28 and FL77-29 exhibited great cytotoxicity in different cancer cells, while HCT 116, Hep G2, and HeLa were more sensitive to both FL77-28 and FL77-29 than FL118. To determine the appropriate concentration range for transport and uptake experiments, the cytotoxicity of FL118, FL77-28, and FL77-29 (0.2, 0.4, 0.6, 0.8, and 1.0 μM) for 4 h on Caco-2 cells was assessed using the MTT assay. As shown in Figure 2, the results showed that the cell viability was above 90%, indicating that there was no significant cytotoxicity in the range from 0.2 to 1.0 μM and the concentrations could be used for the transport and uptake experiments.

### 3.2. Evaluation of Caco-2 Cell Monolayer Integrity

TEER measurement and the permeability of LY were used as the integrity evaluator for the formed Caco-2 monolayer model. The TEER value of more than 300 Ω·cm^2^ and the permeation rate of LY less than 1 × 10^−6^ cm/s indicated the formation of a confluent cell monolayer with well-established tight junctions. In this study, the TEER value was above 600 ± 30 Ω·cm^2^ and the P_app_ of LY was (0.57 ± 0.14) × 10^−6^ cm/s, indicating that the Caco-2 cell model could be used for the transport experiment.

### 3.3. Transport of FL77-28 and FL77-29 across Caco-2 Cell Monolayers

To determine the intestinal transport mechanism and the permeability of compounds, FL77-28 and FL77-29 were added to the AP or BL side of the Caco-2 cell monolayer and incubated for 240 min at 37 °C. The cumulative transported amounts of FL77-28 (Figure 3A) and FL77-29 (Figure 3D) were increased with increasing concentration and the exposure time. The P_app AP-BL_ of FL77-28 (1.0 μM) (Figure 3B) was (2.32 ± 0.04) × 10^−6^ cm/s, and FL77-29 (1.0 μM) (Figure 3E) was (2.48 ± 0.18) × 10^−6^ cm/s. Generally, efflux ratios beyond 2.0 imply that the absorption of the compound may involve efflux transporters [27]. The ER value of FL77-28 (Figure 3C) and FL77-29 (Figure 3F) was 1.14 and 1.40 at 1.0 μM, indicating that further investigation of FL77-28 and FL77-29 transport was still needed to determine by adding inhibitors whether efflux proteins were involved.

### 3.4. Cellular Uptake Kinetic

#### 3.4.1. Effect of Temperature on the Uptake of FL118, FL77-28, and FL77-29

The effect of temperature on the cellar uptake of FL118, FL77-28, and FL77-29 in Caco-2 cells is shown in Figure 4. The uptake of FL118, FL77-28, and FL77-29 at 37 °C or 4 °C was concentration-dependent, increasing approximately linearly with concentration. In addition, there was no significant effect on FL118 (Figure 4A) and FL77-28 (Figure 4B) uptake capacity, indicating that the cellar uptake of FL118 and FL77-28 might depend on passive diffusion in Caco-2 cells. However, incubation at 4 °C significantly increased the cellar uptake of FL77-29 (Figure 4C), which suggested that the absorption of FL77-29 might involve both passive diffusion and efflux pump-mediated mechanisms in Caco-2 cells. Therefore, the efflux pump inhibitors were used to further investigate the cellular uptake of FL77-28 and FL77-29 as follows.

#### 3.4.2. Effects of Transporter Inhibitors on the Uptake of FL118, FL77-28, and FL77-29

To determine whether P-gp and BCRP were responsible for the cellar uptake of FL118, FL77-28, and FL77-29, the Caco-2 cells were pretreated with Verapamil (P-gp inhibitor) or Gefitinib (BCRP inhibitor) for 30 min and then exposed to FL118, FL77-28, and FL77-29 (0.2, 0.4, 0.6, 0.8, and 1.0 μM) at 37 °C for 4 h. These results showed that the uptake of FL118 (Figure 5A,D) and FL77-28 (Figure 5B,E) was not significantly increased in response to Verapamil or Gefitinib, indicating that FL118 and FL77-28 were not substrates of P-gp or BCRP. Gefitinib did not significantly promote the uptake of FL77-29 (Figure 5F), indicating that FL77-29 was not a substrate for BCRP. However, the uptake of FL77-29 (Figure 5C) significantly increased with Verapamil compared to the group without Verapamil, showing that the uptake of FL77-29 was mediated by P-gp.

## 4. Discussion

Given that FL118 was a promising anti-tumor molecule, a series of C7- and C9-substituted compounds using FL118 as the platform were synthesized by our group [23,28,29]. In this work, the results of the MTT assay demonstrated that FL77-28 and FL77-29 were more potent than FL118 in HCT 116, Hep G2, and HeLa cell lines, which was consistent with the MTT results of another C7 fluorinated benzene substituent, 7Q-20, showing superior anti-tumor activity with IC_50_ at the nM level. However, 9Q-20, a C9 substituent was less cytotoxic than FL118, demonstrating that structural modifications at appropriate positions of CPT could affect cytotoxicity.

In addition, the absorption properties of two promising candidates were investigated. Since Caco-2 cells could spontaneously differentiate to form monolayers resembling the intestinal epithelium, the Caco-2 cell monolayer model was chosen. Different from the classic model, the conditions of AP filling with FaSSIF (pH 6.5) and BL filling with HBSS (pH 7.4) were close to the pH of the human small intestine during drug absorption, which better simulated the physiological conditions of the gastrointestinal tract [26]. Moreover, HBSS containing BSA in the BL chamber could provide the driving force for drugs to cross the monolayer. Based on this Caco-2 cell model, the transport of FL77-28 (Figure 3A) and FL77-29 (Figure 3D) showed time and concentration dependence. These preliminary experiments showed that the absorption of FL77-28 and FL77-29 involved passive transport. Due to the P_app_ value serving as a gold standard for assessing the absorption capacity of drugs in vitro, we next calculated the values of these derivatives. Notably, the P_app_
_AP-BL_ values of FL77-28 and FL77-29 ranged from 1.0 × 10^−6^ cm/s to 1.0 × 10^−5^ cm/s at 1.0 μM (Figure 3B,E), implying that these compounds were considered to have moderate absorption according to Biganzoli [30]. However, the Caco-2 model is limited when the drug comes to carrier-mediated routes [31]. Although the ER values of FL77-28 and FL77-29 were less than 2.0 (Figure 3C,F), which seemed to indicate that no efflux proteins were involved, the good permeability of drugs may cover the carrier-mediated pathway, leading to small ER values [32]. Since using ER alone can be misleading when predicting the extent to which efflux proteins impair the absorptive transport of substrates [33], we conducted additional efflux inhibitor experiments.

P-gp can interact with many chemicals, but regarding the mechanism, whether the chemicals are substrates or not is still unknown. The hypothesis of binding to the main binding cavity was recently confirmed by molecular dynamics simulations [34]. Many CPTs undergo P-gp-mediated drug resistance and appear to preferably bind the cavity. However, FL118, a C10,11-substituted CPT, was not a substrate for P-gp, which could be developed as one of the novel strategies to overcome MDR in cancer cells [18]. Therefore, we further investigated whether FL118 derivatives, FL77-28 and FL77-29, were the substrates for P-gp and other efflux proteins. The accumulation amount of FL118 (Figure 4A) and FL77-28 (Figure 4B) was not significantly affected by temperature, but FL77-29 (Figure 4C) was affected. Temperature contributes to energy consumption and the activity of transporter protein. For passive diffusion, temperature also affects the partition/distribution coefficient and passive transcellular (paracellular) permeation [35]. Since low temperature may inactivate the efflux proteins, the uptake of FL77-29 in Caco-2 cells may involve efflux pump-mediated mechanisms, whereas the uptake of FL118 and FL77-28 does not. A further study using efflux protein inhibitors Verapamil and Gefitinib indicated that FL118 (Figure 5A,D) and FL77-28 (Figure 5B,E) were not substrates for P-gp and BCRP. These results again demonstrate that FL118 could bypass some efflux proteins [17,18]. The FL118 derivative FL77-28 also possessed similar excellent properties. On the contrary, the uptake of FL77-29 (Figure 5C,F) was mediated by P-gp, which correlated with the above temperature results. The absorption fate of the other C7-substituted compound, 7-Q20, was mainly dependent on passive diffusion, being pumped by P-gp slightly [28]. The differential abilities of different CPT substitutes to bind to P-gp remain unsolved and require further investigation.

## 5. Conclusions

In the present study, an MTT assay was conducted and the results showed that FL77-28 and FL77-29 possessed better anti-tumor activities than the parent FL118. The transmembrane transport and cellular uptake of FL77-28 and FL77-29 were investigated using Caco-2 cells. The transport of FL77-28 and FL77-29 in the Caco-2 cells was time- and concentration-dependent with the involvement of passive diffusion. The P_app AP-BL_ values of FL77-28 and FL77-29 were above 2 × 10^−6^ cm/s, indicating moderate absorption of these compounds. FL77-29 was a substrate of P-gp. However, FL77-28 bypassed BCRP and P-pg-mediated efflux, suggesting that FL77-28 might be a promising candidate for further clinical development against anti-MDR.

## Figures and Tables

**Figure 1 molecules-27-03669-f001:**
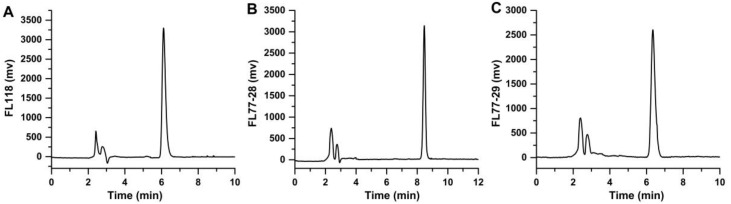
Representative chromatograph, FL118 with a retention time of 6.109 min (**A**), FL77-28 with a retention time of 8.456 min (**B**), and FL77-29 with a retention time of 6.351 min (**C**).

**Figure 2 molecules-27-03669-f002:**
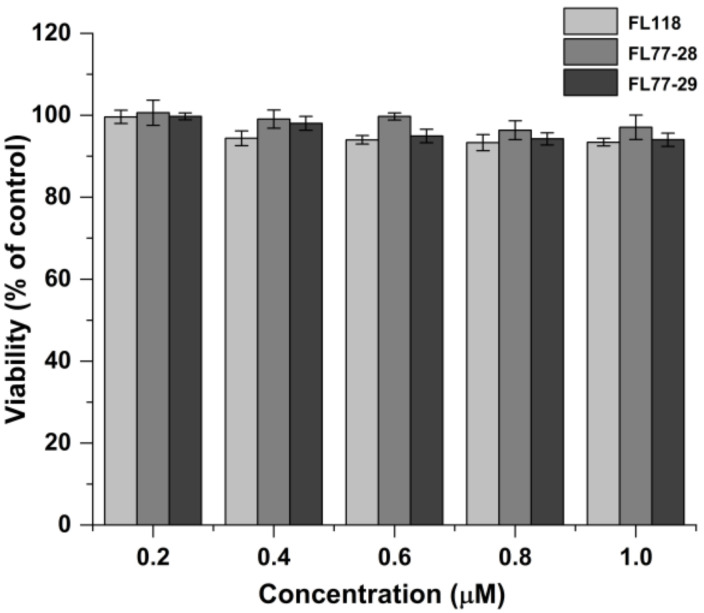
Cytotoxicity of FL118, FL77-28, and FL77-29 for 4 h on Caco-2 cells using MTT assay. Data represent the mean ± SD (*n* = 3).

**Figure 3 molecules-27-03669-f003:**
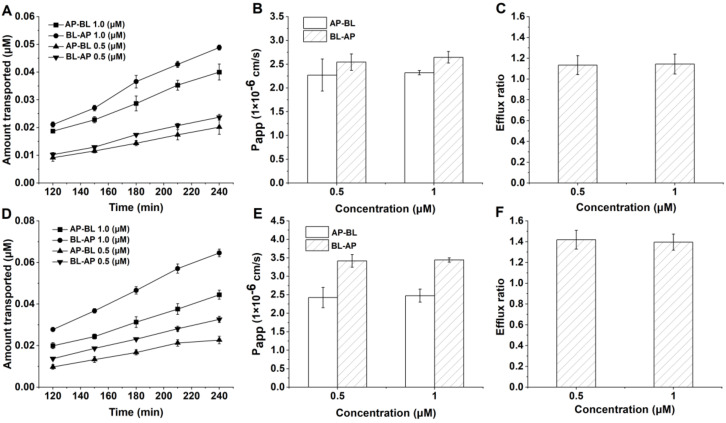
The Caco-2 cell monolayers were incubated with FL77-28 and FL77-29 (0.5 μM or 1 μM) for 4 h at 37 °C. (**A**) Effect of time on the transport of FL77-28 in both directions. (**B**) Permeation studies of FL77-28 on both sides of AP-BL and BL-AP. (**C**) The efflux ratio of FL77-28. (**D**) Effect of time on the transport of FL77-29 in both directions. (**E**) Permeation studies of FL77-29 on both sides of AP-BL and BL-AP. (**F**) The efflux ratio of FL77-29. Results are shown as the mean ± SD (*n* = 3).

**Figure 4 molecules-27-03669-f004:**
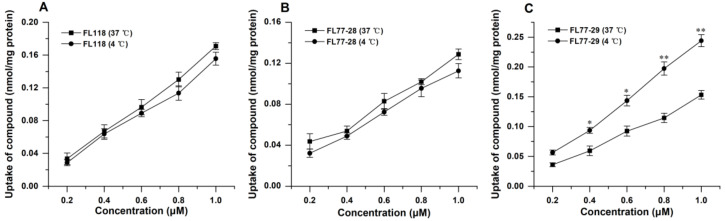
Effect of temperature on the uptake of FL118 (**A**), FL77-28 (**B**), and FL77-29 (**C**) at 4 °C or 37 °C for 4 h in Caco-2 cells. Results are the mean ± SD (*n* = 3). * indicated *p* < 0.05 and ** indicated *p* < 0.01, compared with the results at 4 °C.

**Figure 5 molecules-27-03669-f005:**
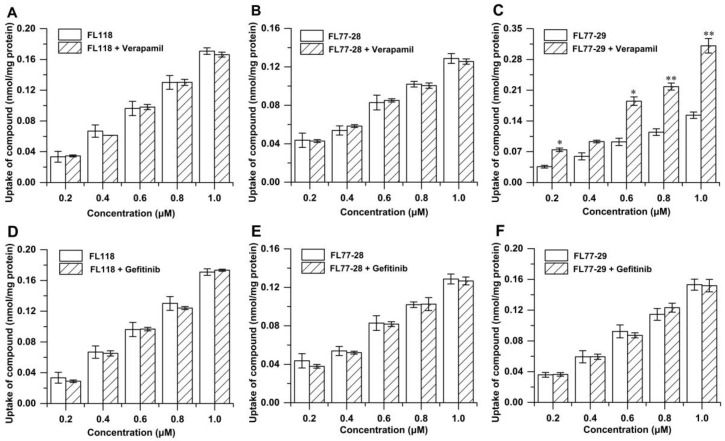
The uptake of FL118, FL77-28 and FL77-29 (0.2, 0.4, 0.6, 0.8, or 1.0 μM) in Caco-2 cell in the presence or absence of P-gp inhibitor-Verapamil (**A**,**C**,**E**) and BCRP inhibitor-Gefitinib (**B**,**D**,**F**) for 4 h at 37 °C. Results are shown as the mean ± SD (*n* = 3). * indicated *p* < 0.05 and ** indicated *p* < 0.01, compared with the control group.

**Table 1 molecules-27-03669-t001:** IC_50_ values of FL118, FL77-28, and FL77-29 against the HCT 116, HepG-2, A549, and HeLa cell lines, determined by MTT assay. Data represent the mean ± SD (*n* = 3).

Compound	HCT 116	Hep G2	A549	HeLa
	IC_50_ (μM)			
FL77-28	0.045 ± 0.014	0.065 ± 0.001	0.048 ± 0.007	0.022 ± 0.007
FL77-29	0.025 ± 0.003	0.034 ± 0.002	0.028 ± 0.004	0.036 ± 0.004
FL118	0.050 ± 0.004	0.104 ± 0.002	0.043 ± 0.004	0.040 ± 0.003

## Data Availability

Data is contained within the article.

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
