# Peer review of "Investigation of the Uptake and Transport of Two Novel Camptothecin Derivatives in Caco-2 Cell Monolayers"

_molecules, 2022, doi:10.3390/molecules27123669_

Round 1

Reviewer 1 Report

The manuscript ID: molecules-1755196 is well written and the aims of the work are well identified

Comment

Line 97 Evaluation of Caco-2 cell monolayer

In the manuscript is not reported the method use to obtain a differentiate Caco-2 cell monolayer. Please include the cell growth condition to rich a monolayer before transport studies.

 Were the cells used - in the 2.6 uptake experiment - at 80% confluence or a complete Caco-2 cell monolayer? The indications must be more precise, and the  TEER indicated.

 Please insert AP and  BL in line 114  for apical and basolateral definition.

 Line 198 specify ER in the test.

 In this work Caco2 cell line was used to determine the appropriate concentration range for transport and uptake experiments. The authors must justify the use of these human epithelial cell originally derived from a colon carcinoma. They used these cells as a model, but Caco2 are an immortalized cell line of human colorectal adenocarcinoma cells, therefore different from normal intestinal cells. Could the transport be different?

The authors should study the effects on transport using normal cells. Please consider the different functionality of normal cells compared to cancerous ones

Authors monitored transport and uptake experiments and cellular monolayer permeability. Besides, intestine-based experiments are totally necessary. To validate the proposed theory and to elucidate the observed outcome, relevant genes should be silenced. This is a preliminary work.

Bibliographic references are quite old, there are no references to recent works (only one from 2019).

Author Response

请参阅附件

Reviewer 2 Report

The authors do a study of the uptake and transport of two novel camptothecin derivatives in Caco-2 cell monolayers. However, I think it cannot be published in the present form.

The following aspects should be taken into account in order to improve the quality of the paper:

1- The abstract needs to be improved

2- Line 32: ATP-binding cassette transporters (ABC transporters)

3- `in vitro´ must be in italic

4- Figures should be improved

5- The synthesized compounds were tested on different cancer cell lines. This work focuses the study on understanding the mechanisms of action in Caco-2 cells. However, only the monolayer model was used. It would be interesting to see these effects in co-culture cell models, for example Caco-2/HT29, as a more realistic model.

6- The discussion must be improved. Other studies should be used to discuss and support the results obtained in this work.

7- A conclusion section is missing in the manuscript

8- The bibliography must be reviewed and recent work must be included in this work.

Round 2

Reviewer 2 Report

The authors made the modifications proposed by the reviewers and the editor. In this sense, in my opinion, the article is ready for publication in the Molecules journal.

Congratulations to the authors!!